# A New Method of Artificial-Intelligence-Based Automatic Identification of Lymphovascular Invasion in Urothelial Carcinomas

**DOI:** 10.3390/diagnostics14040432

**Published:** 2024-02-16

**Authors:** Bogdan Ceachi, Mirela Cioplea, Petronel Mustatea, Julian Gerald Dcruz, Sabina Zurac, Victor Cauni, Cristiana Popp, Cristian Mogodici, Liana Sticlaru, Alexandra Cioroianu, Mihai Busca, Oana Stefan, Irina Tudor, Carmen Dumitru, Alexandra Vilaia, Alexandra Oprisan, Alexandra Bastian, Luciana Nichita

**Affiliations:** 1Department of Pathology, Colentina University Hospital, 21 Stefan Cel Mare Str., Sector 2, 020125 Bucharest, Romania; ceachi.bogdan@gmail.com (B.C.); mirelacioplea@yahoo.com (M.C.); brigaela@yahoo.com (C.P.); cristian.mogodici@zaya.ai (C.M.); liana_ro2004@yahoo.com (L.S.); dragusin_alexandra88@yahoo.com (A.C.); thanatogenesis@gmail.com (M.B.); oana.stefan93@yahoo.com (O.S.); irinafrincu@yahoo.com (I.T.); carmendumitru2004@yahoo.com (C.D.); alexandra.vilaia@gmail.com (A.V.); alexandra.bastian@umfcd.ro (A.B.); luciana.nichita@umfcd.ro (L.N.); 2Zaya Artificial Intelligence, 9A Stefan Cel Mare Str., Voluntari, 077190 Ilfov, Romania; petronel.mustatea@umfcd.ro (P.M.); julian.dcruz@zaya.ai (J.G.D.); 3Faculty of Automatic Control and Computer Science, National University of Science and Technology Politehnica Bucharest, 313 Splaiul Independenţei, Sector 6, 060042 Bucharest, Romania; 4Department of Surgery, University of Medicine and Pharmacy Carol Davila, 37 Dionisie Lupu Str., Sector 1, 020021 Bucharest, Romania; 5Department of Pathology, University of Medicine and Pharmacy Carol Davila, 37 Dionisie Lupu Str., Sector 1, 020021 Bucharest, Romania; alexandra_oprisan@yahoo.com; 6Department of Urology, Colentina University Hospital, 21 Stefan Cel Mare Str., Sector 2, 020125 Bucharest, Romania; 7Department of Neurology, Colentina University Hospital, 21 Stefan Cel Mare Str., Sector 2, 020125 Bucharest, Romania

**Keywords:** artificial intelligence, urothelial carcinoma, lymphovascular invasion

## Abstract

The presence of lymphovascular invasion (LVI) in urothelial carcinoma (UC) is a poor prognostic finding. This is difficult to identify on routine hematoxylin–eosin (H&E)-stained slides, but considering the costs and time required for examination, immunohistochemical stains for the endothelium are not the recommended diagnostic protocol. We developed an AI-based automated method for LVI identification on H&E-stained slides. We selected two separate groups of UC patients with transurethral resection specimens. Group A had 105 patients (100 with UC; 5 with cystitis); group B had 55 patients (all with high-grade UC; D2-40 and CD34 immunohistochemical stains performed on each block). All the group A slides and 52 H&E cases from group B showing LVI using immunohistochemistry were scanned using an Aperio GT450 automatic scanner. We performed a pixel-per-pixel semantic segmentation of selected areas, and we trained InternImage to identify several classes. The DiceCoefficient and Intersection-over-Union scores for LVI detection using our method were 0.77 and 0.52, respectively. The pathologists’ H&E-based evaluation in group B revealed 89.65% specificity, 42.30% sensitivity, 67.27% accuracy, and an F1 score of 0.55, which is much lower than the algorithm’s DCC of 0.77. Our model outlines LVI on H&E-stained-slides more effectively than human examiners; thus, it proves a valuable tool for pathologists.

## 1. Introduction

Bladder cancer is an invalidating and life-threatening disease with a high incidence worldwide. It represents the 10th most common cancer worldwide, with more than 573,000 new cases, and the 13th most common cause of mortality due to cancer, with more than 212,000 deaths in 2020 [1]. Men are afflicted four times more frequently than women, bladder cancer being the sixth most common cancer and the ninth most common cause of death among men worldwide [2]. Five-year survival depends mainly on the tumoral stage; for example, the five-year survival rate in the US is 77.1% (2012–2018), with considerable differences according to the stage: 96% for noninvasive cases, almost 70% for localized (confined to urinary bladder) disease, 40% for regional disease (with regional lymph node affliction), and only 7.7% for metastatic disease [3].

The presence of lymphovascular invasion (LVI) in urothelial carcinoma (UC) is a poor prognostic finding. Both in the transurethral resection of a bladder tumor (TURBT) and radical cystectomy (RC), the presence of LVI is associated with a shorter disease-specific survival rate. Several studies performed on patients that undergo RC showed a significant association between LVI and the different parameters of disease aggressivity, such as the risk of death caused by bladder cancer, risk of recurrence, time to recurrence, overall survival, tumor grade, tumor stage, and lymph node metastasis (Table 1) [4,5,6,7,8]. The studies are quite heterogeneous; they investigate the different parameters associated with LVI treated globally or separately as lymphatic invasion (LI) and vascular invasion (VI). One study finds an association between LI and VI presence [7].

Several studies analyzed LVI in patients who underwent TURBT and found similar results; LVI is associated with disease progression, recurrence, and metastasis [9,10]. 

Some studies identified LVI in UC as an independent prognostic for progression, tumor-free survival disease-specific survival, overall survival, and local and/or distant recurrence [11,12,13]. However, some authors failed to identify it as an independent predictor for recurrence in patients with UC treated using RC [7].

Despite the requirement of including LVI in histopathological reports for UC [14], to date, no consensus has been reached considering the method of identification. Hematoxylin–eosin (H&E) staining cannot properly identify LVI. According to the definition of LVI (emboli), the presence of tumor cells within the lumen of a vessel (either blood or lymphatic vessel) should be demonstrated. Thus, the gold standard for identifying LVI is immunohistochemical (IHC) staining for endothelial markers, i.e., CD31, CD34, D2-40, von-Willebrand factor, or Ulex europaeus agglutinin. The identification of LVI using H&E alone is prone to high levels of error, mostly by under-evaluating its presence [15,16]. Some studies also showed an overdiagnosis of LVI on H&E stain, mostly due to the misinterpretation of retraction artifacts as emboli [17,18].

Ideally, two IHC markers should be performed on each paraffin block of UC; thus, the identification of tumor cells within a vessel is properly recorded as embolus. Considering the costs of this approach (sometimes, one can analyze more than 20 paraffin blocks of a tumor) and the time required for such an analysis, developing an AI-based automated method for LVI identification seems to be a logical step. 

The recent advancements using artificial intelligence have changed the way we approach problems in computational pathology, facing challenges that align closely with the objectives of our research. There have been a few attempts of the automatic histopathologic analysis of UC. Yin et al. developed a machine learning system for distinguishing between the stages of bladder cancer, focusing on nearly 700 features from tumor images to distinguish between noninvasive (stage Ta) and invasive (stage T1) tumors [19]. Similarly, Jansen et al. developed an automated detection and grading system for non-muscle-invasive urothelial cell carcinoma of the bladder by using U-Net segmentation to identify the urothelium within the tissue [20]. Niazi et al. also used U-Net architecture for multi-class image segmentation in order to distinguish various bladder layers from H&E-stained slides [21]. Their success in applying deep learning techniques to these complex problems inspires our approach, showing the potential of segmentation in identifying complex structures within tissue samples.

To our knowledge, no other authors have created a semantic segmented dataset with LVI as the primary target. To accomplish this goal, we aimed to develop a model capable of accurately segmenting and classifying additional classes, like stroma, vessels, invasion, high- and low-grade tumors, nontumor urothelium, and smooth muscle. 

## 2. Materials and Methods

### 2.1. Case Selection

We selected two groups of study: group A for developing the algorithm and group B for the clinical evaluation of LVI presence.

For group A, we analyzed 379 UC cases diagnosed between 1 January 2020 and 31 December 22 (36 months) in the Department of Pathology of Colentina University Hospital in Bucharest, Romania. All the patients had given consent to use their biological material in medical studies. The study was approved by the Ethical Committee of Colentina University Hospital under no. 31/2021.

All the cases were TURBT specimens received by our department as fresh or formalin-fixed tissue; the biological material was submitted for complete histopathologic processing, thus obtaining one or more paraffin blocks per case. The details about tissue processing are available in the Appendix A. 

All the cases were reviewed; cases with small fragments of tissue (less than 8 mm^2^) were excluded. Consultation cases were also excluded. In 5 cases, we selected the slides with inflammatory and/or reactive tissue and less tumoral tissue to provide images of the nontumor urothelium. 

Data were recorded from the histopathologic report (sex, age, the number of paraffin blocks per case, and tumor grade and stage). Only one slide per case was selected. For pale/degraded H&E staining, supplemental sections were cut and stained with H&E in batches of 5 slides on different days to ensure more variable staining. The H&E-stained slides were scanned using an Aperio GT450 automatic scanner (Leica Biosystems, United States); each slide was scanned as a whole slide image (WSI) in “.svs” format, ensuring a uniform image quality. One randomly chosen slide from group A was scanned with a Leica LV1 automatic scanner, and the resulting WSI was used only to normalize the dataset (see Section 2.3).

Finally, we obtained 105 WSIs from 105 different patients (“group A”), used exclusively to select the areas for training, validating and testing the algorithm. These included:One hundred and five cases;One hundred and five slides (all H&E);One hundred and five WSIs (all H&E).

For group B, we selected 55 cases of consecutive cases of invasive high-grade UC, which were also diagnosed using TURBT specimens in our department between 30 June 2022 and 30 June 2023 (12 months). Similarly, we had the patients’ consent and the Ethical Committee of Colentina University Hospital’s approval under no. 31/2021 to use their biological material in medical studies.

The cases included in group A were excluded. The consultation cases were excluded. The cases with immunohistochemical tests for CD34 or D2-40 or other endothelial markers priorly performed for diagnosis were also excluded. Similarly, data were recorded from the histopathologic report (sex, age, the number of paraffin blocks per case, tumor grade and stage, and status of LVI—“H&E LVI”). 

We reviewed all the slides, confirming the invasive character of the tumors; each paraffin block (each of the 55 cases had between one and several paraffin blocks, with a total of 294 blocks) was recut, and consecutive sections were obtained. The first section was stained with H&E, and the next one was stained with D2-40 (clone EPR22182 AbCam, rabbit, dilution 0.5:2000, heat-induced epitope retrieval with EDTA citrate, pH 8) and CD34 (clone QBEnd/10 Leica, mouse, ready-to-use, heat-induced epitope retrieval with EDTA citrate, pH 8). The resulting IHC slides were evaluated for the presence of LVI, “IHC LVI”, as per the gold standard. We did not separately record the emboli present in the lymphatics from those present in the blood vessels. All the slides presenting IHC LVI (H&E, D2-40, and CD34) were scanned with Leica Aperio GT450 automatic scanners (H&E and IHC), and then submitted for annotation. 

Finally, we selected 55 cases of TURBT with a positive diagnosis of invasive UC. In total, there were: Fifty-five cases;Eight hundred and eighty-two slides (two hundred and ninety-four H&E, two hundred and ninety-four CD34, and two hundred and ninety-four D2-40 ones);One hundred and fifty-six WSIs; only the IHC LVI-positive slides (fifty-two H&E, fifty-two CD34, and fifty-two D2-40 ones).

Group B was used for two purposes: a) selecting the areas with LVI for training, validating and testing the algorithm (from 52 H&E slides with proven LVI on IHC stains); and b) analyzing the rate of identification of LVI by pathologists (on all 582 slides from group B).

No patient/case was simultaneously included in both groups.

### 2.2. Annotation Process

The WSIs from group A were annotated by 10 pathologists with various experience levels (Appendix A) using Cytomine (Cytomine Corporation SA, Liège, Belgium). On each WSI, three of the pathologists (with higher expertise in genitourinary pathology) selected a total of 21 rectangular regions of interest (ROIs) larger than 1024 × 1024 pixels (each pathologist selected 7 ROIs on each WSI; the ROIs did not overlap). The ROIs were selected to include representative areas of each WSIs; when rare classes (such as invasion, LVI, muscle and nontumor urothelium) were identified, ROIs were drawn to include them, no matter how small the areas were. The selected ROIs, extracted at 40 × magnification, averaged 1424 pixels in width and 1322 pixels in height. All of the 10 pathologists completely manually annotated the ROIs (“pixel per pixel annotations”) for 21 lesions/structures parameters (further called “classes”): high- and low-grade tumors, adenocarcinomatous differentiation, squamous differentiation, tumor retraction, empty spaces within the tumors, necrosis, invasion, LVI, electrocoagulation, stroma, vessels, inflammation, hemorrhage, smooth muscle, non-tumoral urothelium, normal urothelium, reactive urothelium, von Brunn nests, nondiagnostic, and no tissue. The details about the annotation process are given in the Appendix A.

The WSIs from group B were annotated, starting with LVI. LVI was identified using IHC staining (CD34 and/or D2-40), and then annotated on the H&E-stained WSIs. The ROIs of at least 1024 × 1024 pixels each were constructed around LVI and further annotated using the same procedure described above. The only difference consists in the dimensions of the ROIs; they were highly different in size up to 15,000 × 8000 pixels due to the localization of LVI within the tumor.

Examples of the annotated ROIs are depicted in Appendix A.

### 2.3. Dataset, Image Normalization, and Augmentation Techniques

Smaller samples of 1024 × 1024 pixels were extracted from the ROIs with slight overlaps at the edges to capture comprehensive data from each region. The dataset was divided into training, validation, and testing sets, with distinct WSIs in each, to minimize the bias. The distribution was as follows: 808 samples (57.18%) in the training set, 302 samples (21.37%) in the validation set, and 303 samples (21.44%) in the test set (Figure 1).

To normalize our training data, we employed the StainMixUp method [22]. This technique implies the use of two distinct manually selected image patches, one from the Aperio GT450 scanner (that serves our primary annotated dataset) as our source domain, and another from an auxiliary source, the LV1 automated scanner (Leica Biosystems, 21440 W. Lake Cook Road, Floor 5, Deer Park, IL 60010 United States) as the target domain. This approach allowed us to effectively blend the features from both domains in order to make our model more robust to variations in the staining characteristics between different scanners. By using the StainMixUp method, we mitigated scanner specific biases, leading to a more generalized and accurate model performance across other datasets.

To enhance model generalization, we implemented several augmentation strategies. We focused on Stain variation (using HueSaturationValue and ColorJitter) to simulate the staining variations; texture variation (via ElasticTransform and Sharpen) for tissue texture variability; gentle illumination (via RandomBrightness and FancyPCA) for subtle lighting changes; subtle straining variation (via ColorJitter and RandomToneCurve) to gently alter straining appearance); and tissue section thickness (via RandomBrightnessContrast, GaussianBlur and Sharpen) to replicate the thickness effects. In addition, we also incorporated random flips and 90 degree rotations to simulate various orientations and perspectives.

Delving deeper into our augmentation strategies and reproducibility of our experiments, we utilized the Albumentations library from Pytorch to implement the following techniques:

Stain variation: We used the HueSaturationValue (hue shift limit 20 degrees, saturation shift limit 30%, and value shift limit 20%). Additionally, ColorJitter was applied (brightness 15%, contrast 15%, saturation 15% and hue 15%). These augmentations mimic staining variability, thus enhancing the model’s ability to color inconsistencies.

Texture variation: This was implemented to address the variability in tissue texture arising from sample preparation. Elastic transform (alpha of 2, sigma of 0.1, and alpha affine of 0.1) introduces elasticity in textural representation. Sharpen (alpha between 0.1 and 0.3; lightness between 0.5 and 1.0) accentuates the finer textural details. These methods collectively train the model to recognize diverse tissue morphologies in order to adapt across various sample appearances.

Gentle illumination: This was tailored to introduce nuanced brightness variations, avoiding harsh shadows or overexposure. Random brightness (limit = 0.1) subtly modifies the brightness levels, mimicking the variances in lighting conditions typical in microscopy. FancyPCA (alpha 0.1) adjusts the principal color components, altering tissue perception under varied lighting without drastic changes. These augmentations help the model to adapt to different lighting scenarios, preparing for diverse imaging environments.

Tissue section thickness: This replicates the effects of varying tissue section yhicknesses on the image characteristics. Random brightness contrast (brightness limit 15% and contrast limit 15%) adjusts the brightness and contrast to simulate denser stain absorption in thicker sections. Gaussian blur (blur limit from 3 to 5) introduces a conditional blur effect, representing the slightly out-of-focus quality of the thicker sections. Sharpen (alpha from 30 to 50%) is applied to imitate the clearer edges seen in the thinner sections. These augmentations collectively enhance the model in interpreting sections with varying thicknesses.

All the augmentations techniques were applied to all the images prior to training, ensuring a comprehensive and diverse dataset. During training, we applied normalization and random flips and 90 degree rotations.

### 2.4. Deep Learning Model Development and Training

Our strategy centered around the deployment of InternImage [23]. Unlike traditional CNNs, by incorporating deformable convolutions [24], InternImage is capable of learning various geometric transformations in pose, viewpoint, object scale, and even some deformations, according to the given data. InternImage blends the strengths of CNNs and transformers. Like the capabilities of vision transformers (ViTs) [25], it benefits from large-scale parameters; this becomes a significant advantage when dealing with high variability in histopathological images. 

We used the InternImage-B model with 128 M parameters, using the UperNet method [26] and a batch size of 16 per GPU; the distributed training approach was used to efficiently train, employing 2 A6000 GPU with VRAM of 48 GB per GPU. We trained the distributed model using 2 A6000 GPU with VRAM of 48 GB per GPU, and we adopted the polynomial decay scheduler with a power of 1.0, a warmup ratio of 10^−6^, with a linear warmup of 1500 steps.

By experimenting, we used the AdamW [27] optimizer with a learning rate of 6 × 10^−5^ and a weight decay of 0.05.

Staining standardization was also applied during training as part of the preprocessing using StainMixUp [22], along with a resizing of the images to 512 × 512 pixels using bilinear interpolation, while maintaining the aspect ratio. We trained for 100 epochs and used cross-entropy as the loss function.

### 2.5. Evaluation Metrics

We primarily focused on the dice coefficient (DCC) and intersection over union (IoU) for our model performance; these metrics accurately measure classification and localization in medical imaging contexts [28]. To clinically test LVI identification by pathologists on H&E slides and IHC stains, we used sensitivity, specificity, accuracy, and F1 as performance metrics (Appendix A).

## 3. Results

Group A included 105 patients: 100 cases of UC and 5 cases of cystitis. From the UC cases, 46% represented low-grade UC, and 54% represented high-grade UC; there was male predominance (sex ratio M/F = 2.5:1), and the average age was 66.04 years (interval of 35–89 years). A total of 42% UC cases were noninvasive, while 58% were invasive (28% pT1 and 30% pT2 or above). There were, in total, 260 paraffin blocks (on average, 3.93 paraffin blocks per case) with a range between 1 and 17 paraffin blocks per case. The five patients with cystitis were predominantly female (sex ratio male/female = 1:4), with a median age of 64 years (interval of 53–72 years).

Group B included 55 patients, all of whom had high-grade UC diagnosed using TURBT specimens. Forty-four patients were male (sex ratio is male/female = 4:1), with an average age of 72.49 years (interval of 49–87 years). There were 33 cases (60%) with invasion of the lamina propria (pT1) and 22 cases (40%) with invasion within the muscularis propria (pT2, or at least pT2). Nine cases had one paraffin block, while the others had several blocks per case (up to 23 paraffin blocks), with an average of 5.34 blocks and a total of 294 paraffin blocks.

We obtained 1193 ROIs, from which we selected 1413 samples, each of them including one or more classes. An important aspect of our dataset was class distribution, which exhibited an imbalance, as we can see in Table 2.

### 3.1. Algorithm Results

The results obtained after training the algorithm on the dataset before and after applying the augmentations are depicted in Figure 2. The metrics of the algorithm trained on the non-augmented dataset had DCC values of around 0.7 for most of the priority classes, except the high-grade tumors (0.61), as well as lower values for the IoU, as follows: the high- (0.61 DCC and 0.44 IoU) and low-grade tumors (0.78 DCC and 0.65 IoU), stroma (0.83 DCC and 0.71 IoU), vessels (0.71 DCC and 0.55 IoU), and LVI (0.7 DCC and 0.54 IoU). After implementing the augmentations, the metrics improved by almost 10% for each class, except the stroma, where the metrics remained the same for the following parameters: the high- (0.66 DCC and 0.49 IoU) and low-grade tumors (0.82 DCC and 0.70 IoU), stroma (0.84 DCC and 0.73 IoU), vessels (0.75 DCC and 0.60 IoU) and LVI (0.77 DCC and 0.62 IoU).

For LVI, augmentations led to a noticeable increase in both the dice coefficient and intersection over union scores, from 0.7 DCC and 0.54 IoU to 0.77 DCC and 0.62 IoU, respectively. This is similar for the other critical classes, such as the vessels and high- and low-grade tumors. These results reaffirm the efficacy of the augmentations in refining the model’s performance across various classes.

We analyzed the results of the algorithm related to LVI detection. Most of the cases were properly identified, as we can see in Figure 3. However, both the evaluation metrics showed quite poor results due to the differences in the area of the embolus, as designed by the pathologist and identified by the architecture. In routine practice, a pathologist is concerned with LVI identification (with a result of “present” or “absent”), with the precise area of the embolus being of no value for diagnostic, prediction, or prognosis. In fact, in the first three examples (Figure 3A–C), the algorithm performed superbly. In the last example (Figure 3D), the algorithm correctly identified two of the LVI cases and mistook an area of a tumor with minute peripheral retraction in the periphery as lymphovascular invasion.

Figure 4 lists a series of samples with both Dice and IoU scores of 0 for LVI. Figure 4A depicts a false positive LVI case and a false negative one. When looking at the original image of the sample, the area falsely labeled as LVI on the automated analysis is, in fact, a round small nest of tumors with peripheral retraction and one minute stromal cell lining the artefactual empty space; overall, the appearance closely mimics vascular invasion. One should take into consideration that this image was labeled as an invasive high-grade tumor (and not LVI) based on IHC analysis, proving that the circular space where the tumor lied was not highlighted by either D2-40 or CD34. LVI missed by the algorithm represents a minute fragment of a tumor that is partially present in this sample. Figure 4B–D shows some tumor cells in the empty spaces of tumor retraction falsely labeled as LVI. The tumor retraction is so extensive in Figure 4D that almost any pathologist may over-diagnose LVI in this sample.

Examples of the automatic identification of high-and low-grade tumors, the stroma, vessels, and smooth muscle are included in the Appendix A.

### 3.2. Pathologists’ (Human Examiners) Results

We separately recorded the presence of LVI based on H&E examination alone (H&E LVI) and IHC tests (IHC LVI). Obviously, IHC LVI is the gold standard, certifying the presence of a tumor within the lumen of a vessel (either blood vessel or lymphatic). 

H&E LVI was recorded as reported in the histopathologic reports and revealed LVI in 14 cases (25.45%). 

After examining the IHC stains for D2-40 (294 slides) and CD34 (294 slides), three cases with reported H&E LVI were identified as negative for LVI (false positive cases), while fifteen more cases that had previously been reported as negative presented LVI (false negative) (Table 3). We also recorded the time required to perform this analysis; it varied from 5 min up to 262 min, or 4 h 22 min (the case with 23 paraffin blocks), with a total of 2351 min (39 h and 11 min) and an average of 42.74 min per case.

When compared with the IHC data, the H&E examination had a specificity of 89.65%, a very low sensitivity of 42.30% and an accuracy of 67.27%. The F1 score of H&E human evaluation was 0.55, which is much lower than that of the algorithm (mDCC for LVI 0.77).

The F1 score of human evaluation evaluates the H&E examination results against a ground truth represented by IHC stains. The Dice score of the algorithm evaluates the results of automatic analysis against a ground truth represented by the human annotation of H&E-stained scans doubled by the IHC stains. These findings show that the model is more effective at correctly identifying and outlining LVI using H&E than the pathologists.

## 4. Discussion

The presence of LVI within a tumor is extremely important both for the patients’ prognosis and treatment. The importance of LVI has been studied in various types of cancer; both LI and VI are independent prognostic parameters in endometrial carcinomas [29] and are associated with a decreased survival rate in colorectal cancer [30] and disease-free survival rate in cervical carcinoma [31]. As we have previously shown, many authors identified a direct relationship between the presence of LI, VI, and/or LVI and prognosis, both via univariate and multivariate analyses in urothelial carcinoma [4,5,6,7,8,9,10,11,12,13]. Interestingly, the presence or absence of detrusor muscle on TURBT specimen (which is considered a surrogate marker of resection quality) has no influence on prognosis (recurrence-free survival was not altered by the absence of smooth muscle in the tissue) [32].

However, to date, no method of establishing, identifying, or reporting it has been produced (LI and VI separately or LVI). In fact, as we showed in our study, huge differences occur when LVI is reported based solely on H&E examination and when IHC tests are used. The overall tendency is to under-evaluate LVI, but over-evaluation also occurs. 

We compared our findings with those already reported in the literature, and we found similar results. 

Carlsen et al. evaluated 292 cases of RC for UC; they identified 91 cases with LVI on hematoxylin-azophloxine-saffron slides, while 150 patients had LI and/or BI on IHC. The LVI prevalence (based on IHC identification) was 51.36%, which is similar to that in our study, 47.27% (we had 26 cases with LVI from a total of 55 patients). When looking at the proportion of LVI identification using hematoxylin-azophloxine-saffron alone, the authors over-evaluated in six cases (LVI absent in IHC) and under-evaluated in fifty-nine cases (LVI present in IHC). We calculated the statistical parameters for hematoxylin-azophloxine-saffron LVI identification compared with those of the IHC tests: specificity–95.94%; sensitivity–59.02%; accuracy–77.73%. These findings are not very different to those in our study [16].

Gakis et al. identified LVI in seven more cases using IHC tests in a total of 32 patients with TURBT, and subsequently, RC (accuracy 78.12%) [18].

McQuitty et al. evaluated 22 cases of micropapillary UC. Eight of these cases were negative for LVI on the H&E stains; while using the IHC tests, they identified LVI in seven of these patients. The specificity of H&E-based LVI identification is 100%, the sensitivity is 66.66%, and the accuracy is 68.18% [15].

Other authors did not identify supplementary cases with LVI using IHC staining, but did identify overdiagnosis. Ramani et al. overdiagnosed LVI in three of five cases previously reported as LVI-positive in a study of forty patients with high-grade UC [17].

We re-examined the H&E slides over- and under-evaluated for LVI presence. When the presence of small nests of tumor cells within the lumen of a vessel is identified on H&E-stained slides, the diagnosis of LVI is easy, and there is no need for IHC staining (Figure 5A). The under-evaluated cases had LVI consisting of small groups of tumor cells completely filling the vascular lumina, which were practically indistinguishable from the invasive tumor nests (Figure 5B–D). One case had dozens of emboli revealed only through IHC staining. Thorough analysis revealed some differences between the morphologic appearance of the tumor cells within LVI and the tumor cells from invasive tumor structures; the tumor cells growing in the vessels have more cytoplasm and a more roundish appearance, while the tumor cells from the invasive nests or strands are more compressed, with irregular forms and less cytoplasm (Figure 6A,B). Nevertheless, a pathologist cannot differentiate between an embolus that is growing inside a vessel fully occupying the vascular lumen and an invasive tumor nest based on H&E examination alone; highlighting the endothelial cells bordering the inner aspect of the vessel is mandatory in order to confirm vascular invasion. Over-evaluation occurred in the cases with prominent tumor retraction; in these cases, roundish gaps in the stroma, sometimes bordered by fibrocytes, with groups of tumor cells detached from the stroma, were mistaken as LVI (Figure 6C,D). Over-evaluation occurred in three cases (89.65% specificity), but fifteen cases with LVI detected in IHC were reported as negative using H&E (42.30% sensitivity). 

Considering the importance of LVI for prognosis and therapy, all the authors emphasize the need for LVI reporting; the extremely low detection rate of LVI using H&E compared with its prevalence using IHC stains encourages some authors to recommend IHC assessment on a regular basis. This procedure can be difficult to apply in daily practice. As we showed, we had cases with numerous paraffin blocks–up to 23 blocks. In such a case, to identify LVI, 46 slides stained with CD34 and D2-40 (or 23 slides double-stained with D2-40 and a vascular endothelial cell marker) are needed. The costs are obviously high; moreover, several hours are required for examination, exponentially increasing the pathologist’s workload.

An AI-based algorithm identifies LVI and highlights it on a heat map. The human examiner analyzes the highlighted areas; if the pathologist confirms the presence of LVI, the analysis ends. However, if the nature of the tumor structure is not obviously embolic, the pathologist will select the slide with most numerous areas suspicious for LVI and perform IHC stains on that paraffin block, thus massively reducing the costs and the time required for IHC analysis. 

Despite the promising results, we are aware that several technical issues represent considerable drawbacks in developing a semantic segmentation model robust enough to be used in routine hospital practice. The most important ones are generalization, stain variability, class imbalance, evaluation metric selection, and interobserver variability in annotation.

Generalization is a very important problem. Color variations are a major concern and can affect the results of machine learning models due to the environmental temperatures, scanner type, manufacturer, or stain concentration. Various approaches to handling the domain shift are commonly used, such as stain normalization [33,34,35], color augmentations [36,37] during training, and domain adversarial training [38]. Studies to compare them have also been conducted. For example, during a nuclei segmentation task stain, Lafarge et al. shows how normalization significantly improved the model results (because the model can efficiently learn from the range of staining variabilities observed in those tissue types) [39]. However, color augmentations and staining augmentations lead to better generalization than stain normalization on test sets consisting of unseen tissue types. The authors also suggest that the limitation of stain normalization can be overcome when combined with domain-adversarial training, and thus, it can enable the improved generalization of the learned representation beyond the range of staining distributions seen in the training set (distinguishing between the same and different types of tissue) [39].

Typically, for more robust model training, it is preferable to use WSIs from multiple laboratories to diversify the dataset. Our dataset includes slides from one hospital scanned with one scanner, Aperio GT450 (see Section 2), and these characteristics might affect the generalizability of the AI model.

In our study, to address these challenges, we employed the StainMixUp method [22] with our custom augmentation techniques. For our future studies, we will incorporate data from multiple sources, including different hospitals and scanner types, to ensure the model’s effectiveness across diverse populations and equipment.

Moreover, our dataset shows a significant imbalance in class representation. Classes such as LVI are more underrepresented than others (like the stroma and vessels). This imbalance affects the AI model’s ability to accurately identify the less-represented classes. To address this imbalance, more annotations and augmentations will be employed. However, in real life, there is an imbalance of lesions; for instance, one can observe a single LVI lesion on a slide including 5–6 cm^2^ of tissue; also, invasion can be present in occasional foci on similar large slides. TURBT specimens usually do not include large fragments of smooth muscle, so the presence of smooth muscle in such slides can be scarce.

The selection of evaluation metrics is a critical decision, particularly considering the challenges inherent to the dataset and the nature of the task. We chose DCC and IoU as evaluation metrics because the segmentation labels are not pixel-perfect annotations and can vary in interpretability. In such a case, the DCC offers a more suitable metric because there is discrepancy in the annotations, which is vital in our context, where even the ground truth masks are not perfectly annotated due to human and environmental factors.

DCC calculation combines twice the number of true positive pixels with the sum of the true positives, false positives, and false negatives, giving more weight to the true positives. This approach reduces the penalty for minor misalignments or inconsistencies in annotations, making it a more lenient, yet effective measure of segmentation accuracy. Also, it can better reflect the clinical utility of the model in aiding pathologists, where an exact pixel-level accuracy might be less critical. Moreover, in our future studies, we will include additional evaluation metrics that focus on clinical relevance, such as sensitivity, specificity, and predictive values. These metrics can provide a more comprehensive understanding of the model’s performance in a clinical setting.

Moreover, we also chose our model selection based on the validation performance. Due to the limitation of using samples from a single laboratory and one scanner, and the division of each WSI across datasets, this approach ensures that the model is robust and performs well on unseen data, which is representative of real-world diagnostic scenarios.

Interobserver variability is a very important issue when designing a dataset to train an algorithm [40]. To study the impact of these inherent discrepancies in manual annotation, we previously extensively annotated three cases of UC; eight pathologists with various expertise in UC diagnosis and seniority annotated the same classes used in this study. We identified the two main causes of variability: interpretation (with similarity scores of less than 0.75) and technical problems due to the manual delineation of each area (with similarity scores of approx. 0.9). Interpretation issues (with considerable larger discrepancies between the pathologists) were mitigated in consensus debates to establish a similar approach (i.e., “invasion” category) and similar cut-off levels (“inflammation”, “electrocoagulation”, etc.) [41].

## 5. Conclusions

Lymphovascular invasion identification in urothelial carcinomas is a prognostic and predictive parameter of the utmost importance in diagnosis. Its presence is difficult to establish during routine H&E staining, but due to the costs and time required by examination, IHC staining for endothelial markers is not the recommended diagnostic protocol. We designed the first artificial-intelligence-based automatic method for LVI in UC by analyzing H&E-stained slide scans. Specifically, our algorithm has shown a notable improvement in the identification and segmentation of LVI, which is a critical factor in the diagnosis of UC.

In addition to the model training, our study emphasizes the critical role of dataset preparation, particularly in the annotation process. We adopted a targeted approach by selecting specific ROIs instead of annotating the entire WSI, which proved to be more efficient and effective. Moreover, we established clear rules for segmentation of class assignment in the cases where the annotations were performed by a pathologist and overlapped, thus ensuring consistency and accuracy in our dataset. Such refined annotation strategies are instrumental in enhancing the model’s performance, underlining the importance of dataset quality.

These results not only highlight the potential of AI, but also suggest its utility as a reliable tool for pathologists, particularly in complex diagnostic scenarios where the traditional methods may fall short. This integration promises to improve the diagnostic accuracy, reduce the time taken for analysis, and potentially lead to better patient outcomes; thus, we recommend the automatic detection of LVI as a valuable tool in the hands of pathologists diagnosing UC.

## Figures and Tables

**Figure 1 diagnostics-14-00432-f001:**
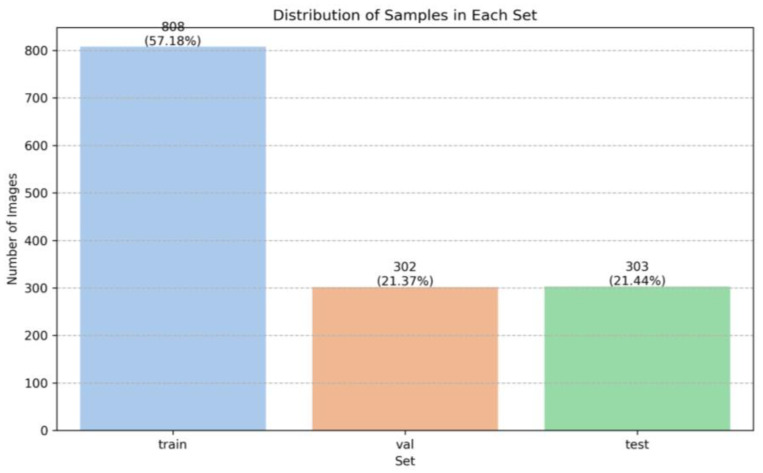
Repartition of samples in the dataset between training, validation, and test sets.

**Figure 2 diagnostics-14-00432-f002:**
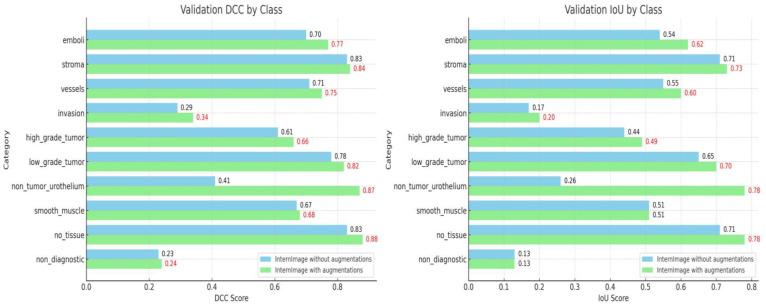
Validation and IoU results per class, with augmentations (columns in blue, DCC and IoU scores in black) and without augmentations (columns in green, DCC and IoU scores in red).

**Figure 3 diagnostics-14-00432-f003:**
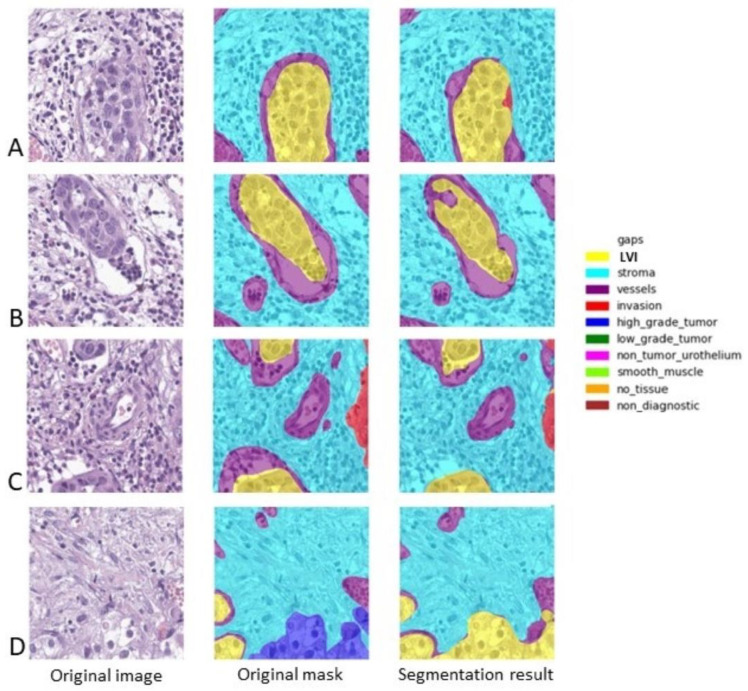
Several examples of automatic identification of LVI. The original sample is depicted on the right, the central sample is the original mask (as it was annotated by the pathologist and further refined in order to eliminate the overlaps), and the right sample depicts the result offered by the algorithm. LVI is depicted in yellow, the stroma is depicted in light blue, the vessels are depicted in purple, high-grade tumors are depicted in dark blue, and invasion is depicted in red. (**A**). Dice score for LVI: 0.92; IoU score for LVI: 0.86; (**B**). Dice score for LVI: 0.87; IoU score for LVI: 0.75; (**C**). Dice score for LVI: 0.82; IoU score for LVI: 0.7; (**D**). Dice score for LVI: 0.32; IoU score for LVI: 0.19.

**Figure 4 diagnostics-14-00432-f004:**
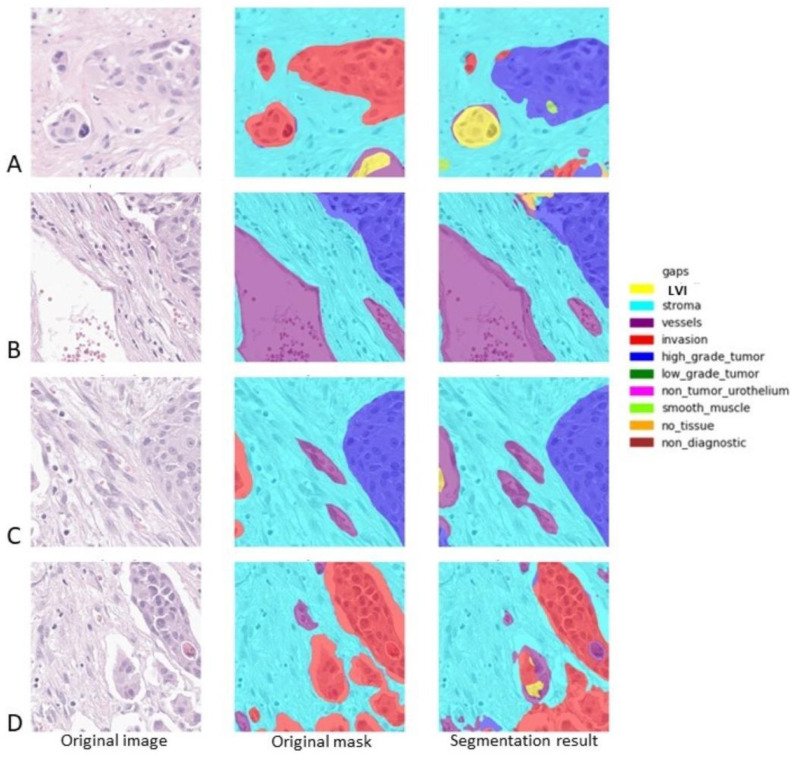
Several examples of automatic identification of LVI. The original sample is depicted on the right, the central sample is the original mask (as it was annotated by the pathologist and further refined in order to eliminate the overlaps), and the right sample depicts the result offered by the algorithm. LVI is depicted in yellow, the stroma is depicted in light blue, the vessels are depicted in purple, high-grade tumors are depicted in dark blue, and invasion is depicted in red. (**A**). Dice score for LVI: 0; IoU score for LVI: 0; (**B**). Dice score for LVI: 0; IoU score for LVI: 0; (**C**). Dice score for LVI: 0; IoU score for LVI: 0; (**D**). Dice score for LVI: 0; IoU score for LVI: 0.

**Figure 5 diagnostics-14-00432-f005:**
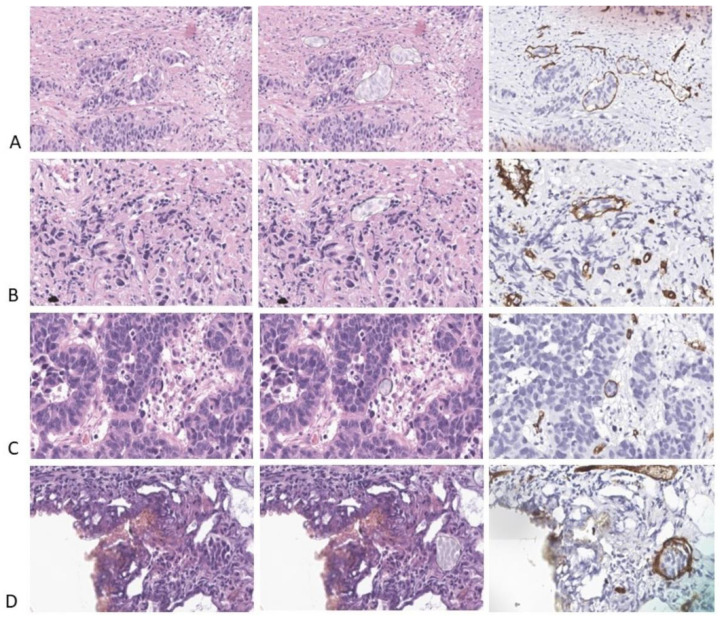
LVI identification by pathologists. (**A**). LVI identified using H&E staining and confirmed on IHC-stained slides for CD34. HE and CD34 20×. (**B**–**D**) LVI underdiagnosed using H&E staining and identified on IHC-stained slides for CD34 (false negative cases). HE and CD34 40×.

**Figure 6 diagnostics-14-00432-f006:**
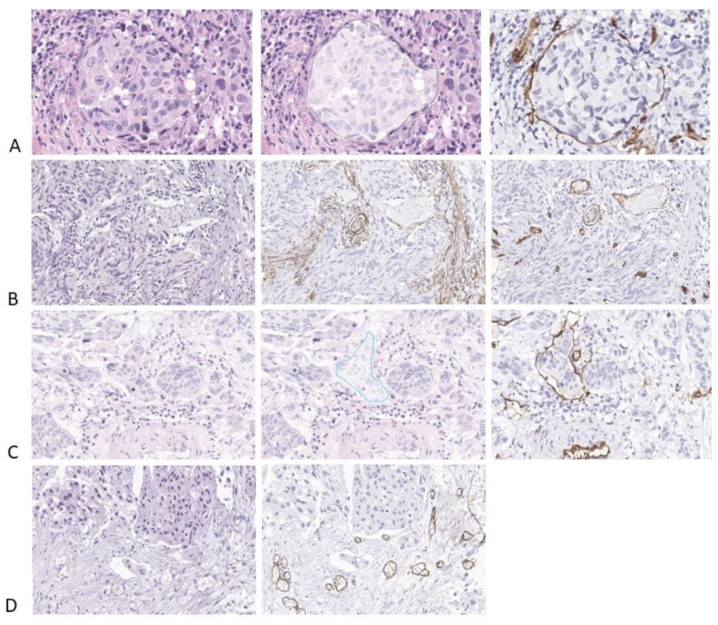
LVI identification by pathologists. (**A**). LVI underdiagnosed on H&E-stained slides and identified on IHC-stained slides for CD34 (false negative cases). The tumor cells of the LVI are larger than the tumor cells from the invasive component. (**A**). H&E and CD34 40×. (**B**). H&E, D2-40 and CD34 × 20. (**C**). Numerous tumor structures exhibit significant retraction in the periphery, which is highly suspicious for lymphovascular invasion; only one structure is a true LVI. H&E and CD34 20×. (**D**). Tumor retraction mimicking LVI; none were confirmed via IHC staining, indicating the overdiagnosis of lymphovascular invasion. H&E and CD34 20×.

**Table 1 diagnostics-14-00432-t001:** Studies on the prognostic significance of lymphovascular invasion.

Study	Cases	Type of Resection	Parameters of Disease Aggressivity	Type of Relationship with LVI
Eisenberg et al. [4]	2403	Radical cystectomy	Risk of bladder cancer death	SSA *
von Rundstedt et al. [5]	499	Radical cystectomy	Time to recurrence Overall survival	SSA
Streeper and al [6]	163	Radical cystectomy	Risk of cancer recurrence Death of disease	SSA
Harada et al. [7]	114	Radical cystectomy	Tumor stage Lymph node metastasis	SSA
Muppa et al. [8]	1504	Radical cystectomy	Cancer specific survival Regional lymph nodes metastasis	SSA
Cho et al. [9]	118	TURBT	Disease progression Metastasis	SSA
Olsson et al. [10]	211	TURBT	Disease progression Recurrence	SSA
Andius et al. [11]	121	TURBT	Progression Disease-specific survival	IPV **
Lotan et al. [12]	958	Radical cystectomy	Local and/or distant recurrence Disease specific survival Overall survival	IPV
Leissner et al. [13]	238	Radical cystectomy	Tumor-free survival	IPV

* Statistically significant association–SSA; ** Independent prognostic value–IPV.

**Table 2 diagnostics-14-00432-t002:** Label frequency distribution per sample in our dataset.

Class	Training Set		Validation Set		Test Set	
	Samples	%	Samples	%	Samples	%
Stroma	714	26.08	262	26.79	272	26.03
Vessels	702	25.64	242	24.74	258	24.69
No tissue	337	12.31	101	10.33	151	14.45
High-grade tumor	294	10.74	82	8.38	90	8.61
Low-grade tumor	237	8.66	97	9.92	114	10.91
Non diagnostic	121	4.42	68	6.95	51	4.88
Nontumor urothelium	103	3.76	3	0.31	64	6.12
Smooth muscle	96	3.51	43	4.4	15	1.44
Invasion	80	2.92	31	3.17	24	2.3
LVI	54	1.97	49	5.01	6	0.57

**Table 3 diagnostics-14-00432-t003:** Correlation between lymphovascular invasion identified in H&E versus IHC by pathologists (human examiners).

	LVI H&E Positive	LVI H&E Negative	Total
LVI IHC Positive	11	15	26
LVI IHC Negative	3	26	29
Total	14	41	55

## Data Availability

Data are contained within the article and Appendix A.

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
