# Peer review of "A New Method of Artificial-Intelligence-Based Automatic Identification of Lymphovascular Invasion in Urothelial Carcinomas"

_diagnostics, 2024, doi:10.3390/diagnostics14040432_

Round 1
Reviewer 1 Report (Previous Reviewer 1)
Comments and Suggestions for Authors
Authors have addressed all my previous questions. Some new comments after reading authors' modifications:
1. Table 1 should be better formatted. In the last column there is no need to repeat "statistical significant" every time. Consider simplification. Plus, it should be cystectomy, not cistectomy.
2. I would encourage the authors to make their dataset public and consider open-sourcing their algorithm as well. It will benefit this field and also make the scientific value of this manuscript stand out.
Comments on the Quality of English LanguageI would recommend extensive English language editing to make it more reader friendly. But this does not affect the soundness of the scientific value of the manuscript.
Author Response
Thank you very much for your thoughtful comments and suggestions regarding our manuscript.
- Table 1 should be better formatted. In the last column there is no need to repeat "statistical significant" every time. Consider simplification. Plus, it should be cystectomy, not cistectomy.
We performed both alterations.
- I would encourage the authors to make their dataset public and consider open-sourcing their algorithm as well. It will benefit this field and also make the scientific value of this manuscript stand out.
We appreciate your encouragement to enhance the scientific value of our work by making our dataset and algorithm publicly available. We share your enthusiasm for advancing the field through open science practices. However, after careful consideration, we regret to inform you that we are unable to comply with the request to make our dataset public or open-source our algorithm at this time. The datasets presented in our article are not readily available for several reasons. Primarily, the data are part of a commercial product patent request, which imposes certain restrictions on their distribution. Additionally, the data were obtained from a third party, further limiting our ability to make them available due to contractual obligations and privacy concerns.
Reviewer 2 Report (New Reviewer)
Comments and Suggestions for Authors
The aim of the study was to to develop a model capable of accurately segmenting and classifying additional classes like: stroma, vessels, invasion, high- and low-grade tumors, nontumor urothelium and smooth muscle in UC. Topic is interesting and study properly performed. However, there are a few critical aspects that Authors should review in order to improve the overall quality of manuscript.
- Sensitivity and specificity of test could be reported as well as AUC, in order to suppose s practical application in clinical settings.
- Absence / presence of detrusor muscle on TURBt specimen could be integrated. (PMID: 32900676).
Author Response
Thank you very much for your effort and your thoughtful comments and suggestions regarding our manuscript.
The aim of the study was to develop a model capable of accurately segmenting and classifying additional classes like: stroma, vessels, invasion, high- and low-grade tumors, nontumor urothelium and smooth muscle in UC. Topic is interesting and study properly performed. However, there are a few critical aspects that Authors should review in order to improve the overall quality of manuscript.
- Sensitivity and specificity of test could be reported as well as AUC, in order to suppose s practical application in clinical settings.
We calculated the sensitivity and specificity of the test in which we establish the level of identification of lymphovascular invasion by pathologists (human examiners) – table 3. The data from this table are not suitable to calculate AUC.
- Absence / presence of detrusor muscle on TURBt specimen could be integrated. (PMID: 32900676).
We include the impact of presence of detrusor muscle in TURBT specimen, thank you very much for your suggestion.
This manuscript is a resubmission of an earlier submission. The following is a list of the peer review reports and author responses from that submission.
Round 1
Reviewer 1 Report
Comments and Suggestions for Authors
The scientific questions raised in this paper is important and the methodology descirbed in the paper is detailed. I would recommend the following points to make it better.
Introduction:
1. Page 2, Line 56-63 - it would be better to talk through these points instead of using outlines
2. Page 3, Line 107 - changing "had to" to "aimed to" should be better
Method:
3. The rationale for choosing two-groups method are not 100% clear
4. Method is a bit redundant, I would recommend simplying it, especially 2.1 - 2.3
5. Inter-annotator relaibility should be reported
Results:
6. I would recommend limit the example figures to shorten the results.
Discussion:
7. Limitations should be more straightforward and should be at the end of the discussion.
Comments on the Quality of English LanguageLanguage quality is good. I would recommend simplifying sentences to make it more reader friendly.
Reviewer 2 Report
Comments and Suggestions for Authors
The manuscript's structure currently presents challenges in terms of coherence and logical flow, making it difficult for readers to follow the presented arguments and methodologies. Additionally, the information provided regarding the methods is insufficient for reproducing the experiments. The introduction of non-standard terminology such as 'lymphovascular invasion EMBOLI' is not advisable. The term 'lymphovascular invasion' is more precise and recommended for use. It is essential for this work to enhance its clarity and conciseness to meet the standards required for consideration in the peer review process.
Comments on the Quality of English LanguageFormatting and syntax issues have been identified and noted. The phrase "golden standard" should be corrected to "gold standard."